The impact of land reform on the status of large carnivores in Zimbabwe

http://orcid.org/0000-0001-7205-7426 Williams Samual T. 1 2 3 s.t.williams@durham.ac.uk
Williams Kathryn S. 1 2
Joubert Christoffel J. 4
http://orcid.org/0000-0002-7601-5802 Hill Russell A. 1 2
1 Department of Anthropology, Durham University , Durham , United Kingdom
2 Primate and Predator Project , Lajuma Research Centre , Limpopo Province , South Africa
3 Dambari Wildlife Trust , Bulawayo , Zimbabwe
4 Selati Game Reserve , Gravelotte , Limpopo Province , South Africa
Pimm Stuart
Electronic publication date: 2016 Jan 14
Publication date: 2016
Volume: 4
Electronic Location ID: e1537
Received 2015 Jul 25; Accepted 2015 Dec 6
Copyright: © 2016 Williams et al.
Copyright year: 2016
Copyright holder: Williams et al.
License: This is an open access article distributed under the terms of the Creative Commons Attribution License, which permits unrestricted use, distribution, reproduction and adaptation in any medium and for any purpose provided that it is properly attributed. For attribution, the original author(s), title, publication source (PeerJ) and either DOI or URL of the article must be cited.
License URL: https://creativecommons.org/licenses/by/4.0/

Keywords: Zimbabwe, Africa, Land reform, Carnivore, Spoor, Resettlement

Funding: This work was funded by the Howard G. Buffett Foundation, Sea World Busch Gardens, the Whitley Wildlife Conservation Trust, Marwell Wildlife, Colchester Zoo, the Durham University Department of Anthropology, and St Mary’s College, Durham University. The funders had no role in study design, data collection and analysis, decision to publish, or preparation of the manuscript.

==============================
Large carnivores are decreasing in number due to growing pressure from an expanding human population. It is increasingly recognised that state-protected conservation areas are unlikely to be sufficient to protect viable populations of large carnivores, and that private land will be central to conservation efforts. In 2000, a fast-track land reform programme (FTLRP) was initiated in Zimbabwe, ostensibly to redress the racial imbalance in land ownership, but which also had the potential to break up large areas of carnivore habitat on private land. To date, research has focused on the impact of the FTLRP process on the different human communities, while impacts on wildlife have been overlooked. Here we provide the first systematic assessment of the impact of the FTLRP on the status of large carnivores. Spoor counts were conducted across private, resettled and communal land use types in order to estimate the abundance of large carnivores, and to determine how this had been affected by land reform. The density of carnivore spoor differed significantly between land use types, and was lower on resettlement land than on private land, suggesting that the resettlement process has resulted in a substantial decline in carnivore abundance. Habitat loss and high levels of poaching in and around resettlement areas are the most likely causes. The FTLRP resulted in the large-scale conversion of land that was used sustainably and productively for wildlife into unsustainable, unproductive agricultural land uses. We recommended that models of land reform should consider the type of land available, that existing expertise in land management should be retained where possible, and that resettlement programmes should be carefully planned in order to minimise the impacts on wildlife and on people.

Introduction

Large-bodied mammals of the order Carnivora (hereafter referred to as large carnivores) are culturally important to humans; their body parts are used in ceremonies and traditional medicine and they feature in storytelling, mythology and witchcraft (Kruuk, 2002). Large carnivores are depicted in artworks, on currencies, on coats of arms and on the kits of sport teams (Loveridge et al., 2010). They provide important ecosystem services such as helping to maintain wildlife abundance and richness, and enhancing carbon storage (Ripple et al., 2014). They can also bring in large revenues through tourism (Barnes, 2001; Lindsey et al., 2007) and hunting (Jorge et al., 2013; Lindsey et al., 2006), but they can be a financial burden through predation on livestock (Rust & Marker, 2014).

Despite their value, large carnivores across the world are in decline (along with their prey: Ripple et al., 2015) as a result of the growing human population and increasing pressures on the environment (Di Marco et al., 2014; Gittleman, Macdonald & Wayne, 2001; Nowell & Jackson, 1996; Ray, Hunter & Zigouris, 2005; Woodroffe, 2000), and they are particularly vulnerable to anthropogenic disturbance (Ray, Hunter & Zigouris, 2005; Sillero-Zubiri & Laurenson, 2001). Many protected areas have failed to sufficiently protect large mammals from anthropogenic threats (Craigie et al., 2010; Lindsey et al., 2014), and the persistence of national parks alone may not be sufficient to safeguard even species that are relatively abundant in protected areas (Child, 2009a). The importance of land outside of state-protected areas to biodiversity conservation is therefore becoming increasingly clear (Bond et al., 2004; Fjeldså et al., 2004; Kent & Hill, 2013).

Large-scale privately owned land is often much more extensive than state protected areas and generally has a relatively low human population density (de Villiers, 2003; du P. Bothma, Suich & Spenceley, 2009; Lindsey et al., 2013a; Lindsey et al., 2013b; Odendaal, 2006; Scoones et al., 2010), so is capable of supporting relatively large wildlife populations (Child, 2009c; Lindsey et al., 2013b). For example, before 2000, 30% of the land area of Zimbabwe was composed of large-scale private farms (20% of which were managed specifically for wildlife), while state protected reserves occupied just 13% of the country (Table 1; du Toit, 2004; Scoones et al., 2010). As a result, private land supported substantial wildlife populations, including 80% of the cheetahs (Acinonyx jubatus) in Zimbabwe (Stuart & Wilson, 1988). Other species such as wild dog (Lycaon pictus) and brown hyaena (Parahyaena brunnea), which, like cheetahs, are outcompeted by larger carnivores in national parks (Durant, 1998; Mills, 1990; Woodroffe & Ginsberg, 2005), also occurred in relatively large numbers on private land in Zimbabwe and other countries (Creel & Creel, 1996; Kent & Hill, 2013; Pole, 2000; Stuart & Wilson, 1988).

Table 1 Land distribution in Zimbabwe immediately before the onset of the FTLRP (2000) and in May 2010.

Adapted from Scoones et al. (2010).

Land use type	2000	2010	
	Area (million ha)	% of total land area	Area (million ha)	% of total land area	
Large-scale private farms	11.7	29.9	3.4	8.7	
Small-scale private farms	1.4	3.6	1.4	3.6	
Old resettlement (1980–2000)	3.5	9.0	3.5	9.0	
New resettlement (2000–present)	0.0	0.0	7.6	19.5	
Communal land	16.4	41.9	16.4	41.9	
National parks and forest land	5.1	13.0	5.1	13.0	
Other land	1.0	2.6	1.7	4.3	
Total	39.1	100	39.1	100	

Much of the prime agricultural land in Zimbabwe was alienated by the colonial administration and gazetted as private land, leaving much of the poorer quality land as communal land (Kwashirai, 2009; Wels, 2003). At independence in 1980, communal land made up 41.9% of Zimbabwe’s land area, and was settled by Africans who largely practiced subsistence agriculture (Scoones et al., 2010). In contrast, Zimbabweans of European descent (an ethnic minority) owned almost all of the large-scale private land, which comprised 36.6% of the land area, and was used primarily for commercial agriculture (Scoones et al., 2010). Since independence in 1980, efforts have been made in Zimbabwe to redress the racial imbalance in land tenure. Progress, however, had been slow (Clover & Eriksen, 2009), partly because the commercial farms on private land were highly productive, enhancing food security and providing employment for approximately a third of the Zimbabwean workforce (Kwashirai, 2009; Magaramombe, 2010). Between 1980 and 2000, resettlement occurred through a relatively organised process, with the government purchasing available properties on a willing-seller, willing-buyer basis, or later by compulsory acquisition (Spierenburg, 2011). Criteria for resettlement included underutilisation, absentee or multiple ownership of properties, and proximity to communal areas.

In 2000, Zimbabwe entered the fast-track phase of its land reform programme, whereby private land was redistributed to African settlers, often taken by force and without payment of compensation for the land (Cliffe et al., 2011; Hughes, 2010). While some observers portrayed this as a grassroots movement, many others contended that this was organised by the government in order to destabilise the perceived support base for the opposition party (Chari, 2013; Willems, 2004; Zunga, 2003). This resulted in haphazard resettlement of large areas of private land (Table 1), most of which was then utilised for subsistence agriculture by communities (Scoones et al., 2010). The new farmers cleared much of their land, but many lacked the resources, support, experience or training necessary to farm effectively (DeGeorges & Reilly, 2007; Fakarayi et al., 2015; Scoones et al., 2010). The impacts of this violent process on socio-economic factors has been well documented (Chimhowu & Hulme, 2006; Cliffe et al., 2011; Kapp, 2009; Kinsey, 2004; Magaramombe, 2010; Waterloos & Rutherford, 2004), but despite the great potential for impacting on wildlife, there have been no systematic studies of the impacts of land reform on the status of wildlife (Purchase et al., 2007; Williams, 2007).

This study uses the partial resettlement of Savé Valley Conservancy (SVC; Fig. 1) in south east Zimbabwe as a case study to determine the impact of land reform on the status of cheetah, leopard (Panthera pardus), lion (Panthera leo), wild dog, brown hyaena, and spotted hyaena (Crocuta crocuta). The impact that land reform had on the status of large carnivores in SVC between 2000 and 2008 is then evaluated through an assessment of the population sizes of large carnivores in private, fast-track resettlement (hereafter referred to as resettlement) and communal land use types (LUTs).

Figure 1 Land use types and spoor transects conducted at the study site in 2008.

An old resettlement area (settled in 1982) also shared a boundary with SVC, but was not included in this study as it predated the FTLRP (Zinyama, Campbell & Matiza, 1990). A total of 1,036 km of transects were sampled. Inset map shows the location of Savé Valley Conservancy in relation to Gonarezhou, Kruger and Limpopo National Parks and national boundaries.

Materials and Methods

The study area was made up of three LUTs in south-eastern Zimbabwe (central coordinates 20°22′S and 31°56′E): private, resettlement and communal. The private LUT study area was the Savé Valley Conservancy (SVC), a private game reserve that originally covered approximately 3,490 km2 (Fig. 1), constituting 10.3% of the remaining private land in Zimbabwe. SVC was established from former cattle ranches as a cooperatively managed wildlife area (Lindsey et al., 2009), a process catalysed by the reintroduction of black rhinoceros (Diceros bicornis) as part of the government’s conservation strategy, the difficulties of farming livestock in such a drought-prone area and the greater profitability of wildlife in relation to cattle in semi-arid environments (Child, 2009b; Lindsey et al., 2009; Price Waterhouse, 1994). Trophy hunting became the main economic activity in SVC (Lindsey et al., 2009), as previously successful ecotourism proved unviable after the collapse of Zimbabwe’s tourist industry due to the civil unrest associated with the FTLRP (Mkono, 2012).

In 2000 and 2001 an area of SVC measuring 960 km2 was resettled as part of the FTLRP, reducing the area of SVC to 2,530 km2. The criteria for selection of the properties for resettlement were not transparent (Chaumba, Scoones & Wolmer, 2003) and there were no apparent differences between the properties that were resettled and their neighbours that were not resettled in terms of habitat, rainfall, or the density of wildlife perceived by the landowners before resettlement (Williams, 2011). The communal LUT study area was made up of an area of 715 km2 of communal land located to the west of SVC. To the south of SVC private game reserves and private farms link the study site to Gonarezhou National Park and the Greater Limpopo Transfrontier Park. Communal lands make up most of the remainder of the borders of SVC.

The topography of the region is gently undulating, with gneiss, Para gneiss and granite outcrops rising up to 250 m above ground (Pole, 2000), and an elevation of 480–620 m above sea level (Pole et al., 2004). Soil quality is poor and rainfall is low (474–540 mm per annum) and highly variable, with a wet season between November and March and a dry season between April and October (Lindsey et al., 2009; Pole et al., 2004). The main vegetation type is deciduous woodland savannah, with Colophospermum mopane, Acacia tortillas and Acacia-Combretum woodlands, and riparian vegetation along the watercourses (Pole et al., 2004). The study site falls into the Zambezian and mopane woodlands ecoregion (Olson et al., 2001).

Spoor counts were conducted in October and November 2008 along existing gravel roads. Spoor counts are a widely used method of estimating the density and abundance of carnivores (Balme, Hunter & Slotow, 2009; Bauer et al., 2014; Boast & Houser, 2012; Crooks, 2002; Deryabina et al., 2015; Fritz et al., 2003; Funston, 2001; Groom, Funston & Mandisodza, 2014; Gusset & Burgener, 2005; Houser, Somers & Boast, 2009; Johnson et al., 2010), and can provide robust estimates across a wide variety of species and a broad geographical range (Funston et al., 2010; Midlane et al., 2015). Roads on which spoor were sampled were generally composed of substrates that preserved spoor well such as hard sand (Stuart & Stuart, 2003). A vehicle was driven at a steady speed of 20 km/h in the early morning (generally between 05:00 and 08:00), following Stander (1998). An experienced tracker sat on the front of the vehicle while scanning the transect for spoor, and stopping the vehicle to examine any spoor of large carnivores encountered. Transects were driven towards the sun where possible in order to facilitate the detection and identification of spoor (Liebenberg, Louw & Elbroch, 2010). The species, number of individuals and location of each spoor was recorded. Spoor were disregarded if they were over 24 hours old or if the spoor were thought to be from an individual that had been recorded earlier on the transect that day, which was determined from spoor morphology, group size and direction of travel (Bauer et al., 2014; Funston et al., 2010; Stander, 1998).

The relationship between spoor frequency (the number of kilometres of transect driven between records of spoor of a particular species) and sampling effort (the number of spoor recorded) was investigated through bootstrap analyses on inter-spoor intervals (the distance between each spoor observation for a particular species, when transects are systematically combined). This was conducted by calculating 95% confidence intervals from two randomly sampled inter-spoor intervals with replacement, then progressively increasing the sample size and calculating fresh confidence intervals with each sample (after Stander, 1998) using R version 3.2.0 (R Development Core Team, 2015). The code used for bootstrap analysis is available from Williams (2015a). This made it possible to determine whether sufficient data had been collected to reach the preferred levels of variation and sampling precision (Stander, 1998).

Carnivore spoor density is correlated with population density (Funston et al., 2010). Spoor density at the study site was used to estimate the population density and size of carnivores at SVC by applying the models developed by Stander (1998) (see Williams (2011) for a discussion of model selection). This applies a linear function to spoor density to calculate population density, using calibration data from study sites with known spoor densities and population densities of study animals.

The raw data analysed in this study is available in Williams (2015b). The research had approval from both the Durham University Department of Anthropology Departmental Ethics Committee, and the Durham University Life Sciences Ethical Review Process Committee.

Results

Across 1,036 km of transects, a total of 65 lion, 101 leopard, 10 cheetah, 129 wild dog, 12 brown hyaena and 106 spotted hyaena spoor were collected. Sample penetration (the ratio of sum of transect lengths (km) to survey area (km2)) for most LUTs was close to the value of 7 recommended for these techniques (Stander, 1998) (Table 2). Bootstrap analyses on transects in the private LUT (in which almost all spoor were recorded) showed that variation in spoor frequency stabilized at approximately 30 spoor for lion, leopard, cheetah, brown hyaena and spotted hyaena, and at 60 spoor for wild dogs (Fig. 2). These sample sizes were not met for cheetah or brown hyaena spoor, resulting in large confidence intervals for these species. Sampling precision initially increased sharply, but changed little after 30 spoor for lion, leopard and spotted hyaena (Fig. 3; mean change 15% between 30 spoor and 65 spoor, the minimum sample size collected for these species). Sample sizes for cheetah and wild dog spoor were too small for sampling precision to stabilize. The desired level of precision and variation was therefore reached for most species. Estimation of population size was still conducted for all species but levels of variation and sampling precision were taken into account in interpretation of the results.

Figure 2 The relationship between spoor frequency and sampling effort for large carnivores on transects on private land at Savé Valley Conservancy in 2008.

Circles represent means and lines represent 95% confidence intervals. Spoor sample size was 65 for lion, 101 for leopard, 10 for cheetah, 129 for wild dog, 12 for brown hyaena and 106 for spotted hyaena.

Figure 3 The relationship between coefficient of variance and sample size for large carnivores on transects in private land on Savé Valley Conservancy in 2008.

Spoor sample size was 65 for lion, 101 for leopard, 10 for cheetah, 129 for wild dog, 12 for brown hyaena and 106 for spotted hyaena.

Table 2 Areas of each land use type in and around Savé Valley Conservancy, and survey effort of spoor counts conducted in 2008 to determine the spoor density of large carnivores and other mammals.

Land use type	Area (km2)	Sum of transects (km)	Sample penetration	Total length surveyed (km)	
Private	2,530	346	7.3	696a	
Resettlement	960	149	6.5	149	
Communal	984	110	8.9	110	
Total	4,474	605		955	
Note:

aPrivate transects were each sampled twice in order to increase the sample size. On resettlement and communal land transects were sampled only once as there were too few spoor recorded to make this necessary.

Spoor from all large carnivore species were recorded in the private LUT, while only spoor from spotted hyaenas were detected in the resettlement LUT, and no large carnivore spoor were recorded in the communal LUT (Table 3). Spoor densities (defined as the number of carnivore spoor per 100 km of transect) differed significantly between land use types (Kruskal-Wallis: χ2 = 14.087, df = 2, P = 0.01), and were greater in the private LUT than the resettlement and communal LUTs (Table 3). The private LUT was estimated to support 11 cheetah, 193 leopard, 72 lion, 142 wild dog, 114 spotted hyaena and 13 brown hyaena (Table 3). In contrast in the resettlement LUT only 6 spotted hyaena were estimated to occur, while the communal LUT supported no large carnivores (Table 3).

Table 3 Population size and population density estimates for large carnivores across each LUT in and around Savé Valley Conservancy in 2008.

Values in parentheses represent 95% confidence intervals. Stander’s (1998) leopard equation was used to calculate the estimates for the leopard, while (Stander’s, 1998) lion and wild dog equation was used to calculate the estimates for all other species (see Williams, 2011).

	Population density (animals/100 km2)	Population size	
Species	Private	Resettlement	Communal	Private	Resettlement	Communal	
Cheetah	0.44 (0.41)	0	0	11 (10)	0	0	
Leopard	7.64 (1.73)	0	0	193 (44)	0	0	
Lion	2.85 (1.17)	0	0	72 (30)	0	0	
Wild dog	5.65 (3.19)	0	0	143 (81)	0	0	
Spotted hyaena	4.51 (1.05)	0.61 (0.44)	0	114 (27)	6 (4)	0	
Brown hyaena	0.53 (0.39)	0	0	13 (10)	0	0	

Discussion

In 2000 and 2001, approximately 40% of SVC was resettled as part of the FTLRP. In 2008, large carnivore densities in the remaining private LUT were comparable to those found in protected areas elsewhere (Bailey, 2005; Bauer & Van Der Merwe, 2004; Ivan, White & Shenk, 2013; Mills & Hofer, 1998; Thorn et al., 2009; Woodroffe, McNutt & Mills, 2004). In contrast, carnivores occurred at very low densities or were absent in the resettlement areas and communal LUT.

Although there are no comparable density estimates from before resettlement, it seems unlikely that the patterns we report were due to low population densities in the resettlement areas prior to resettlement. Sighting frequencies of cheetah on Senuko ranch declined markedly following the onset of the FTLRP and resettlement on other properties in SVC (Williams, 2011), and carrying capacity estimates for large carnivores based on the biomass of potential prey species from aerial surveys decreased between 2004 and 2008 (Williams, 2011). Similarly, while animal populations could respond to resettlement through changes in behaviour between the different LUTs, reducing group size and use of roads (and thus spoor frequency) (Stillfried et al., 2015), this should not influence prey biomass estimates and carrying capacity estimates from aerial surveys. A difference in the population density of large carnivores between LUTs resulting directly from resettlement is the most likely explanation for our results.

The absence or low densities of large carnivores in the resettlement and communal LUTs can be explained by high human densities, which led to pressure for land to grow crops and graze livestock, resulting in a loss of habitat and prey base. In the private LUT human population density was low, habitat was still comparatively intact and prey was relatively abundant. Even so, carnivore population sizes appear to have been below carrying capacity estimates based on prey availability and rainfall (Williams, 2011), although this may have been partially do to the fact that carnivore populations were still thought to be recovering from their low densities before SVC was formed (Lindsey et al., 2009).

The low carnivore densities in the resettlement LUT are most likely the result of a population decline in response to the resettlement process, rather than migration of animals out of resettlement areas. If this were the case, we would expect to find greater densities of wildlife on private land near to resettlement areas, but the opposite trend was observed (Williams, 2011). No evidence was found of carnivore populations moving from the resettlement areas to the communal land surrounding SVC. A more likely explanation is population declines precipitated by extensive bushmeat poaching (Lindsey et al., 2011b).

The extremely high levels of poaching in SVC were the result of a large human population being settled on private land with large wildlife populations, and were exacerbated by Zimbabwe’s economic crisis and food shortages arising from the FTLRP (Knapp, 2012; Lindsey et al., 2011a; Moss, 2007), limiting carnivore abundance in the private LUT. Poaching rates in SVC increased to extremely high levels after the FTLRP began; between August 2001 and June 2009 over 84,000 snares were removed and 4,148 poachers were captured (Lindsey et al., 2011b). The remains of 6,454 poached animals were recovered, including 2 cheetahs, 5 leopards and, 27 wild dogs (Lindsey et al., 2011b). Numerous individuals of prey species were also recovered during this period, such as 2,606 impala (Lindsey et al., 2011b), which would reduce carnivore carrying capacity through removal of the prey base (Hayward, O’Brien & Kerley, 2007). Within the private LUT, rates of poaching per unit area were over 2.5 times higher in the south than the north (Lindsey et al., 2011b), which is probably linked to greater proximity to the resettlement area (Fig. 1). When resettlement occurred the perimeter game fencing was stolen, facilitating access of poachers from the resettlement area to southern SVC and providing abundant material to manufacture snares (Lindsey et al., 2009). While fencing can be an incredibly useful tool for managing wildlife populations (Packer et al., 2013), it is important to use material that cannot be easily used to manufacture snares (such as Veldspan™ or Bonnox™), rather than the steel and barbed wire that was used to construct the fence at SVC (Lindsey et al., 2012).

Within SVC land resettlement has thus had a large impact on large carnivore populations. Land resettlement was widespread in Zimbabwe, however, and most of the other large-scale conservancies including Gwayi, Bubiana and Chiredzi River have also been severely affected by the FTLRP (du Toit, 2004; Lindsey et al., 2011b), with very few (such as Malilangwe Trust) remaining untouched (Lindsey et al., 2011b). In addition to conservancies, almost all other private land was resettled, so if the trends at SVC are indicative of trends across Zimbabwe, this could have severe impacts on the status of large carnivores. While a small proportion of resettled land may have been retained for wildlife-based uses, a preliminary extrapolation of our findings suggests that Zimbabwe’s FTLRP could have had a significant negative impact on the population size of large carnivores at a national scale, resulting in estimated population declines of an average of 36%, up to a maximum of 70%, across the country, depending on the species (Article S1). Species that depend on private land to a greater extent, such as cheetah, are likely to have been more strongly affected than species such as lions, whose populations are concentrated in protected areas. This combination of potential steep population declines and disrupted connectivity throughout the Greater Limpopo Transfrontier Park, brought about by resettlement removing corridors and links between national parks, calls into question the viability of the remaining populations of some species in Zimbabwe; relatively large populations of up to several thousand individuals are thought to be required in order to maintain genetic viability (Crooks, 2002; Lande, 1995). In addition to affecting wildlife populations, the FTLRP is likely to have resulted in wide scale loss of the jobs (Lindsey et al., 2013a; Lindsey, Roulet & Romañach, 2007), community benefits (Le Bel et al., 2013), food security (Cumming, 2005) and income through tourism (Naidoo et al., in press) or hunting (Lindsey et al., 2006) associated with the wildlife industry.

A key factor that enabled the wildlife industry to become so important and the wildlife populations to become so abundant on private land in Zimbabwe and other countries in southern Africa, was the introduction of legislation devolving rights to utilise wildlife on private land to the landowners (Bond et al., 2004). This allowed landowners to exploit a ready market of photographic tourists (Naidoo et al., in press) and trophy hunters (Lindsey et al., 2006; Lindsey, Roulet & Romañach, 2007), while encouraging landowners to manage their land to maximise wildlife populations, leading to significant growth in the occupancy of wildlife populations (Child, 2009b). In the semi-arid areas in which most land managed for wildlife occurred, wildlife was the most appropriate land use in terms of economic productivity (Child, 2009b), employment (Bond et al., 2004), and environmental conservation (Bond et al., 2004), and rain-fed agriculture was not recommended (Vincent & Hack, 1960). The FTLRP ignored the reasons for the shift from agriculture to wildlife and resulted in the replacement of viable wildlife operations with unsuitable farming practices. While the beneficiaries of the FTLRP did accrue benefits such as access to land and natural resources (Scoones et al., 2010), this came at great cost to both society and biodiversity conservation.

The negative impacts of land reform on the status of large carnivores documented here could be reduced by modifying the way in which land reform programmes are implemented. Firstly, the model of land reform that was applied under Zimbabwe’s FTLRP considered agricultural models at the expense of a wildlife-based model. The agricultural land reform models applied were poorly suited to the arid and semi-arid areas in which many private wildlife and livestock ranches were located (Child, 1995; Vincent & Hack, 1960), and when combined with poor availability of resources for the new farmers this contributed to crop failure (DeGeorges & Reilly, 2007). If a wildlife-based land reform model could be applied, whereby private wildlife ranches retain wildlife as a land use but a more representative ethnic profile of landowners is achieved, this could result in stronger wildlife populations, be more ecologically sustainable, provide greater profits (Child et al., 2012; Price Waterhouse, 1994) and lead to lower levels of human-wildlife conflict (Williams, 2011). It appears that this has started to happen, changing the way in which the government addresses land reform (Scoones et al., 2012), but care must be taken to ensure that this is done in a sustainable way.

Planning is critical to minimising the impact of land reform on wildlife and human-wildlife conflict. Many problems could be avoided by considering wildlife when planning land reform, such as by maintaining connectivity between wildlife populations (Bennett, 2003) and reducing edge effects by minimising the boundary between resettlement and wildlife areas (Balme, Slotow & Hunter, 2010; Woodroffe & Ginsberg, 1998). Where resettlement has already fragmented habitats (du Toit, 2004), wildlife corridors could be re-established to link separated populations and enhance their viability (Bennett, 2003). Any wildlife remaining in the areas of resettlement land that became reincorporated into SVC as wildlife corridors could be owned by the communities resettled in the area and jointly managed by the community members and SVC. Funds raised through utilisation of this wildlife resource could go back to the community, enabling them to benefit from conserving wildlife on their land.

Allowing local communities to benefit economically from the wildlife in SVC, for example through schemes like CAMPFIRE (Frost & Bond, 2008; Taylor, 2009a; Taylor, 2009b), would create an incentive for them to protect wildlife populations in the area and reduce the need for people to turn to poaching (Campbell, 2000). Indeed this is now happening; for example, a trust has been established to purchase wildlife breeding stock on behalf of the neighbouring communities to be placed in SVC (Kreuter, Peel & Warner, 2010). The offspring are sold to SVC, providing a regular income to the communities.

Other innovative mechanisms for involving communities in conservation on private land have been explored in South Africa. For example, game reserves such as Phinda and Mala Mala were claimed by communities, who then leased the land back to the reserve mangement, maintaining wildlife as the land use and retaining the expertise and capital of the former owners, but bringing revenue to the community (Masombuka, 2015; Spenceley & Rylance, 2012). Similar programmes have also been successful in national parks. Sections of Kruger National Park and the Kgalagadi Transfrontier Park in South Africa have been claimed by communities, who were granted legal ownership of the land. The communities now manage the land under a contractual agreement with the government, and retain the rights to commercial development such as tourist lodges (Grossman & Holden, 2009). Raising funds to allow communities to buy shareholdings in SVC would enhance community participation in the conservancy and allow them to benefit either through paying dividends to community members or by funding community projects such as schools, clinics or irrigation projects (Taylor, 2009a). Another option is to expand private reserves to include community land. This has been undertaken at SVC, whereby 25 km2 of cattle grazing land was set aside and became part of the conservancy (Lindsey et al., 2009). Partnerships between communities and the private sector such as these could provide a more durable land use model than the largely exclusive ownership of extensive areas of land by a minority ethnic group, and models such as these may prove to be a sustainable solution to the land reform issue. If authorities could provide greater security of land tenure to beneficiaries of the FTLRP, attitudes towards wildlife may become more positive (Romañach, Lindsey & Woodroffe, 2007), which could also lead to reduced rates of poaching (Hartter & Goldman 2011).

We suggest that further research is conducted to determine that the trends observed at the study site are representative at national and international levels, and whether carnivore populations in Zimbabwe are continuing to decrease further. Land reform initiatives are also underway in other countries that had extensive areas of private land such as South Africa and Namibia. Before land reform programmes were initiated (de Villiers, 2003; Kepe, Wynberg & Ellis, 2005; Lahiff, 2014), private land constituted 72% and 44% of the total land area of South Africa and Namibia, respectively (Adams & Howell, 2001). The pace of redistribution, however, has again been slow, with only approximately 1% of private land in South Africa and Namibia being redistributed by 2000 (Adams & Howell, 2001), prompting some stakeholders to call for a more radical approach such as the Zimbabwean model of land reform (de Villiers, 2003; O’Laughlin et al., 2013). With land reform remaining an important issue around the world (Adam, 2013; Diniz et al., 2013; Nyahunzvi, 2014; Pellegrini & Dasgupta, 2011; Vilpoux, 2014), the recommendations of this study could help to prevent the socio-economic and wildlife issues that Zimbabwe has encountered from being repeated elsewhere.

Conclusions

Land reform appears to have significantly reduced the population size of large carnivores in SVC. Very high levels of poaching and a decline in prey base associated with land reform are thought to be responsible for these declines. This case study could be indicative of broader trends across Zimbabwe. We recommended that care is taken to carefully plan land reform programmes in other countries in order to minimise the negative effects on wildlife populations and maintain linkages where possible. Retaining wildlife as a land use, while employing innovative models that retaining existing expertise and capital, would go a long way towards allowing both wildlife and people to benefit from land reform.

Supplemental Information

Supplemental Information 1 A preliminary assessment of national population trends of large carnivores in Zimbabwe.

Click here for additional data file.

The African Wildlife Conservation Fund very kindly allowed the spoor data they collected to be used towards this analysis. We thank the authorities that granted permission to conduct the research and to the participants for taking part in the study. We would also like to thank the landowners, managers, and workers of Savé Valley Conservancy, particularly on Chishakwe and Humani, for their hospitality and support. Many thanks to Rueben Bote and Misheck Matari for doing an amazing job as trackers and to Phil Stephens for writing the R script used for bootstrapping. Finally, we are grateful to three anonymous reviewers for providing very valuable comments and strengthening the manuscript.

Additional Information and Declarations

Competing Interests

Author Contributions

Animal Ethics

Data Deposition

The authors declare that they have no competing interests. Christoffel J Joubert is an employee of the Selati Game Reserve.

Samual T Williams conceived and designed the experiments, performed the experiments, analyzed the data, wrote the paper, prepared figures and/or tables, reviewed drafts of the paper.

Kathryn S Williams performed the experiments, wrote the paper, reviewed drafts of the paper.

Christoffel J Joubert conceived and designed the experiments, performed the experiments, reviewed drafts of the paper.

Russell A Hill conceived and designed the experiments, analyzed the data, wrote the paper, reviewed drafts of the paper.

The following information was supplied relating to ethical approvals (i.e., approving body and any reference numbers):

The research has approval from both the Department of Anthropology Departmental Ethics Committee, and the Life Sciences Ethical Review Process Committee, both at Durham University.

The following information was supplied regarding data availability:

The code used for bootstrapping was uploaded to GitHub: Williams ST (2015a) Code for analysing spoor data. DOI: 10.5281/zenodo.20991. Available from https://github.com/samual-williams/spoor-analysis.

The raw data used in the manuscript was uploaded to Open Science Framework: Williams ST (2015b) Spoor data. Open Science Framework. DOI 10.17605/OSF.IO/EHAZX. Available from https://osf.io/ehazx.

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
