# Peer review of "The impact of land reform on the status of large carnivores in Zimbabwe"

_PeerJ, doi:10.7717/peerj.1537_

## Round 0.1 · original submission · Major Revisions

Both the reviewers have made important comments, so please address them carefully. Reviewer 1 notes that vital data are not readily accessible to the reader. They must be, even if you have to append the paper mentioned to your revision by way of support during the review process.

As always with these matters, the more promptly you return your revision, the more willing the reviewers will be to reassess it.

Reviewer 1 ·

Basic reporting

The MS is well-written with sufficient background material and references. The Figures and Tables are also sufficient. However, some of the data is not available for review, it is apparently located in William 2015b. This impedes my ability to review the current MS adequately.

Experimental design

The MS is original and the topic relevant and important. While the field methods appear to have been conducted thoroughly and adequately, I cannot accept the population extrapolations. The methodology to extrapolate population numbers from a single study site to the entire country is not justified.

Validity of the findings

I disagree with the validity of the findings. The conclusions they draw for SVC may be appropriate. However, the findings of greater importance (the proposed overall decline in population numbers across Zimbabwe) are not properly justified or evidence-based. It is highly speculative. A MS that dealt strictly with the effects of land reform on the region surrounding SVC would be much more appropriate rather than attempting to extrapolate across the country. Said extrapolation could still be included in a MS on SVC however I envision it more as a cautionary paragraph rather than central Results material.

Additional comments

L101. Confusing. Was the SVC itself resettled? When was the formation of SVC? More description of the area can go in Materials but this is still confusing. Is it just “centered” on SVC?
L109. More on SVC needed. History? Does the SVC include all three land use types? Was there any particular reason/ what was the choice behind resettlement of some of the land? Was it more/less productive land? Earlier you set up that private land was generally prime agricultural land but that doesn’t seem to be the case here… Why is part of the communal land sampled noncontiguous with the other parts? What separates it?
L661. Table 2. Why was the commercial length doubled? Sum of transects is only 346 but total length becomes 696? This is not true for resettlement or communal. Could use more explanation in the methods.
The data from 2000 that allows the determination of trends is not shown or discussed in detail. Proving this data is of high quality and substantially comparable to the data collected in the study would help the MS.

Reviewer 2 ·

Basic reporting

The paper is interesting, but it would be more useful if it gave less weight to some of the trends -- and considered potential confounds in interpreting the results.

Experimental design

Besides the land redistribution program, Zimbabwe has had significant economic problems the past few years -- how can we be sure the changes in carnivore populations sizes result from the land policy or from the impacts of abject poverty?

Validity of the findings

The reported trends are given too much weight -- although the authors stress that some of the data are too sparse to provide accurate estimates, this caveat is largely forgotten in the discussion. Also, it would be worth employing a Bayesian analysis of the populations trends, since this would provide a better indication of the uncertainties inherent in such short-term time-series data.

---

## Round 0.2 · Major Revisions

I was able to find a reviewer on this revision who has a substantial knowledge of conditions in Zimbabwe. (S)he has raised many concerns not noticed by earlier reviewers. Nonetheless, attending to these concerns will surely improve this manuscript considerably.

Reviewer 1 ·

Basic reporting

This 2nd submission is an improvement over the earlier work and the authors addressed the primary concerns with the previous MS.

Experimental design

In this 2nd submission, experimental design is adequate for the conclusions drawn.

Validity of the findings

The countrywide extrapolations have been removed and the conclusions now better match the analysis.

Additional comments

Were there any prior, and how recent, density estimates for large carnivores in SVC? How do your results fit within that context?
Second, the main focus of L118 is that there were no apparent differences between properties that got resettled. Briefly, how and what comparisons were made?
Third, in the Discussion, a paragraph should be added describing the uncertainty in the analysis. For instance, it is not conclusively proven that carnivore densities were not different between these areas before the 2008 spoor surveys were undertaken. Additionally, there are other possible reasons why carnivore spoor did not show up in the communal areas, such as changes in behavior and smaller group sizes of carnivores in unprotected landscapes.

Reviewer 3 ·

Basic reporting

Review of Williams et al.
This paper is valuable because it represents one of the first documentations of the severely negative impacts on wildlife that were associated with land ‘reform’ in Zimbabwe. As such, the paper would be a valuable addition to the literature.
However, before publishing, the paper requires better framing of the issues. For example:
a) The paper needs better framing around the issues pertaining to land reform and wildlife in Zimbabwe. For example, because it was driven largely as a political imperative for the incumbent government to retain power (they were losing popularity rapidly and turned to land redistribution as a means of patronage) – practical considerations went out of the window – which is why land that was being used productively, such as SVC was converted into completely unproductive land uses – such as subsistence agriculture... resulting in a lose-lose scenario – where the wildlife was lost, the jobs associated with that wildlife were lost, where future potential community benefits from wildlife were foreclosed, and where the recipients of the land are highly food insecure and poor with no immediate prospects of escaping that trap.
b) There is need for better background on why wildlife developed as a land use in Zimbabwe pre land reform – the combination of legislative change allowing landowners user rights over wildlife, coupled with a ready market for hunting and photographic tourism, coupled with the fact that in the low lying and low rainfall parts of the country – rain-fed agriculture and livestock production were shown to be unviable. Wildlife populations increased on private land hugely – in the context of large conservancies such as SVC (and several others), as isolated game ranches, on mixed wildlife-livestock ranches, and on game sections of crop farms. Cumming 2004 in Brian Childs Performance of Parks book is a good source, as is a paper by Ivan Bond. The fast track land reform programme ignored the reasons for the shift to wildlife – and encouraged the practicing of unsuitable land uses on dry land.
c) While the fast track land reform programme conferred benefits to the recipients in terms of access to land, firewood, and wildlife – it came at the cost of the jobs and livelihoods of the people employed on the ranches, the economic productivity of those ranches and also the economic benefits associated with wildlife-based land uses. It was also deeply unfair and prejudicial to the landowners who lost out, and their workers.
d) The paper needs to shed light on what happened in other large scale conservancies in Zimbabwe – providing a short review from the literature and perhaps discussions with experts (e.g. Gwayi – taken over by politically connected elites and run down due to lack of investment and expertise; bubiana and chriedzi river – largely resettled with small scale farmers and wildlife virtually obliterated; SVC – partially re-settled, Malilangwe/Bubye – not settled and wildlife continuing to perform well. There are insights from papers by Lindsey et al. 2011 on the impacts of land reform on wildlife, and a review conducted by Raoul du Toit in 1994 on the impacts
e) The paper needs to shed light on how the zim government eventually realised that the resettlement of wildlife lands with subsistence farmers was resulting in the loss of the wildlife resource – resulting in the development of the ‘wildlife-based land reform policy’ in about 2008 or so – which slowed the rate of such resettlements and changed the way in which they addressed land reform.
f) The paper needs to show an understanding of current events and approaches related to land reform in SVC – e.g. SVC is now trying to partner with communities – which potentially provides a long term solution to the issue of land reform – while achieving sustainability and retention of the only really viable land use in that area.
g) The paper needs to shed light on successful models for wildlife-based land reform that have been implemented elsewhere in the region. Examples include Makuleke in Kruger, Phinda, Mala Mala in South Africa – there ownership was transferred to communities, but the existing expertise and capital was retained and wildlife remained the land use – so they were win wins. Whereas the SVC model has been a lose-lose. Basically the authors should clearly outline why the method of achieving land ‘reform’ in SVC was a disaster – and offer alternatives on how it could be done – e.g. buying shareholdings for communities on existing wildlife ranches, land use planning to clear some areas of the resettled zones for wildlife as community wildlife conservancies (also retaining connectivity within SVC) and extending the boundaries of SVC out to include community lands – etc.
h) The resettlement within and around SVC threatens connectivity within the greater Limpopo TFCA

Specific comments
Scoones has produced some incredibly spurious statements and findings about land reform in Zimbabwe and the ‘success’ thereof.
In the abstract – mention how the land reform resulted in a shift from a sustainable and productive land use (wildlife) in a semi-arid area to an unsustainable and unproductive land use – so in future there is a need for models of land reform that retain the most productive land use for the type of land in question – which in the case of semi-arid savannahs is often wildlife.
Line 46 – their prey is also declining rapidly – consider quoting Ripple’s herbivore decline paper
Correct spelling of hyena is hyaena, savanna – savannah, recognize - recognise
Line 75 Zimbabweans of European descent
Line 78 – partly because the commercial farms were highly productive, ensured food security and employed large numbers of people (benefits which were all lost during the land reform process).
Line 84 – this is where you have to mention the political nature of the land reform – point (a) above
Line 85 – taken by force without compensation (and with associated displacement of hundreds of thousands of workers)
Program should be spelled programme
Line 110 – and because of the comparative profitability of wildlife – a bit of background needed here on why wildlife turned out to be a more productive land use in that environment than livestock
Line 113 - though significant potential for the development of ecotourism exists when the political and economic situation turns one day.
Line 208 – in addition to massive bushmeat poaching. The eradication of wildlife on resettled lands in Zimbabwe is indicative of a widespread, but relatively poorly acknowledged, bush meat crisis in African savannahs. It is like a chronic disease that emerges during times of duress. Authors should acknowledge this manifestation of the bush meat crisis.
Line 2011 – this is because predator numbers were still increasing – the conservancy was only formed in 1991 – predators were not reintroduced (except for ten lions) – and so the recovery took time. The predator densities are much higher now in the privately owned areas than they were when the authors did their field work. However – that said poaching is a limiting factor – and is also the reason why wildlife is so suppressed on the resettled areas (along with habitat loss).
Line 225 – you might want to allude to the fencing debate here (e.g. Packer et al. 2013) – this case emphasizes the need for fencing to be made out of material that cannot be made into snares – such as bonnox / veldspan – not the steel and barbed wire that was used to make the SVC fence.
Line 230 – to clarify – large areas were allocated for communal resettlement, but some were reallocated to private individuals. In the latter scenario there is scope for retention of wildlife-based land uses – even though in many cases wildlife in such areas (e.g. Gwayi) has taken a major knock because: a) the new guys often don’t have the expertise to run wildlife; and b) they have not invested in protection of the resource. However the potential remains.
Line 234- certainly this is congruent with Esther van der meer’s work on cheetahs – she has found that cheetahs have been largely extirpated from private lands in Zim – outside of a couple of conservancies still functioning as wildlife areas.
Line 249 – in the past in Zimbabwe land has been categorised into ‘natural regions’ based on their sitability for agriculture. SVC is in natural region 5- meaning it is dry and highly drought prone. This means that rain fed agriculture is unlikely to succeed. In addition – the landowners that were there before tried and failed to run commercial livestock – in spite of having benefits such as solid veterinary input and the ability to afford to provide supplementary feed in the event of a drought
Natural region 5 is barely suitable for extensive livestock production – and as the commercial farmers found out, is much more suited to wildlife production than anything else. The benefits of wildlife production in that very dry and harsh environment (characterised by high variability in annual rainfall – which makes livestock production so hard – and which caused the commercial ranchers to almost go bust in the last major drought in 1991) are:
a) Wildlife production is decoupled to a large extent from primary productivity – because the value comes from a tourism experience – rather than a simple linear relationship between grass and meat production
b) Because wildlife make use of more varied components of the primary productivity
c) Because wildlife are better adapted to the climate than livestock
d) Because of the high demand for hunting and tourism experiences

What we have now is a situation where:
a) The new farmers in SVC fail to produce enough food to meet their needs resulting in a reliance on food aid, and in some cases on maize imported from Zimbabwean farmers who were evicted from their land in Zim and who are now farming in Zambia
b) Livestock production that is destined to fail in the same way that it did when it was run commercially
c) Very high levels of human wildlife conflict – crops, livestock and impacts on human life
d) Foregone scope for wildlife production due to encroachment of land and poaching of wildlife
e) The economic activity and jobs that were produced on the wildlife ranches that were taken over has been lost. In future, the scope for photographic tourism in SVC is very high, and tourism produces a lot of jobs

The way forward for SVC and other areas now is to rationalise land uses – through a process of negotiation whereby portions of the re-settled land area earmarked for wildlife production, which are owned by the communities and jointly managed by the communities and SVC. This kind of public private partnership could create a situation where out of the ‘ashes’ comes a much more robust and sustainable land use model than the one before where the land was all owned by people of a minority ehnic group. Then for the remaining wildlife land in the conservancy, a possible solution to achieve greater community participation would be to seek funds to purchase shareholdings for the communities in SVC.
Positively, along the western border of SVC there are large areas of relatively lightly populated land that could be incorporated into SVC as community owned wildlife land under joint management
During land reform with wildlife – there is a case for trying to retain some of the existing expertise, rather than simply booting them out
Recommendations for land reform in dry areas where wildlife was the primary land use:
a) Try to retain wildlife as the land use – as that was chosen by the previous owners because it was the most viable option
b) Try to retain some of the existing expertise and capital (as lack of both has been part of the problem)
c) Learn from some of the novel models that have been attempted successfully in the southern African region that have achieved both a and b
Line 272 – ‘We suggest’ is better than ‘it is suggested’
Line 275 – be specific – Namibia and South Africa, which also have large wildlife ranching industries in lands that are often unsuitable for small scale subsistence agriculture
Line 281 – aside from the problems associated with the change in land use in Zim, land reform initiated a massive economic collapse in the country as a whole – so one would hope they do not go down that route.

Experimental design

The experimental design is fine - though the authors would do well to review the literature to shed light on whether the impacts they documented are widespread

Validity of the findings

Perhaps the authors, to avoid criticism - could indicate how sure are they that the patterns they observed are not simply due to wildlife having moved out of the resettled areas (rather than being poached en masse)

---

## Round 0.3 · accepted · Accept

Thank you for your prompt attention to the reviewers' comments. I have not sent them back to the most critical reviewer, but feel that you have done what's needed to answer them.

Let me stress that PeerJ does not extensively edit manuscripts. What you write is what will appear. The Word version has numerous green underlines, suggesting that you could improve the grammar. In any case, I urge you to read your text most carefully!